# The Income Equalization System among Municipalities in Norway: Strengths and Implications

**Johannes Idsø [1]** 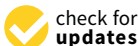**, Torbjørn Årethun [1] and Bharat P. Bhatta [2],***

[1]   Institute of Business and Administration, Western Norway University of Applied Sciences, 6856 Sogndal, Norway; johannes.idso@hvl.no (J.I.); torbjorn.arethun@hvl.no (T.Å.)

[2]   Center for Research and Development, King's College, Babarmahal, Kathmandu 44600, Nepal

*   Correspondence: johannes.idso@hvl.no; Tel.: +977-47-5767-6304

**Abstract:**  Norway is a leading nation pursuing egalitarian public policies. With an aim to smooth out income differences between municipalities and to stabilize individual municipality's revenue over time, Norway has implemented a scheme called the income equalization system among municipalities. The scheme, which transfers revenue to poor municipalities, helps to maintain similar welfare services in all municipalities. We present and illustrate the model with data from all municipalities in Norway. We also demonstrate how the scheme helps stabilize tax revenue across municipalities over time. Furthermore, we show how the scheme can cause poor municipalities to have reduced incentives to increase their tax revenue.

**Keywords:** income equalization system; tax revenue of municipalities; efficiency and equality; Norway

**JEL Classification:** H71

## 1. Introduction

Equality of wealth, income and opportunity has always received much attention from scholars and decision-makers in a public policy worldwide (Corak 2013; Mankiw 2011; Stiglitz 2012; Blank 2002; World Bank 2006). Many countries therefore pursue egalitarian policies. As part of the egalitarian policies, Norway has been implementing a scheme called the income equalization system (IES) among municipalities for decades with an aim of maintaining similar welfare services in all municipalities irrespective of their revenues (Idsø and Årethun 2013). The IES involves transfer of some tax revenue from richer municipalities to poorer ones. Economists, on the other hand, generally assume that it is not possible to achieve both equality and efficiency because equality-obtaining measures essentially result in a loss of efficiency—economic efficiency in particular.

Despite extensive literature dealing with the efficiency costs of efforts/measures of income transfers from the rich to the poor, researchers have paid little attention to the social cost of inequality, the benefits of the equality of income, opportunity in the long term and measures to minimize the costs and augment the benefits; this is because social efficiency is at the cost of economic efficiency. It is generally accepted that if there is an opportunity to transfer a small fraction of our income to someone who needs it more than us, we should not hesitate. The basic philosophy underlying the IES is the same. This issue of the IES necessarily involves political philosophy not only economics. Discussion of inequality necessarily involves our social and political values and normative judgement. Public policy scholars must understand this.

Both equality and efficiency are important issues in a policy decision. They are also equally vital in an economic analysis. Economists often assume that there is a trade-off between the two (Okun 1975). Economists are also blamed for emphasizing efficiency, economic efficiency in particular, at the cost

of equality (Stiglitz 2012). However, Adam Smith himself believes that "No society can surely be flourishing and happy, of which the greater part of the numbers are poor and miserable. It is but equity, besides, that they who feed, clothe and lodge the whole body of the people, should have such a share of the produce of their own labor as to be themselves tolerably well fed, cloathed, and lodged" (Smith 1976).

A highly influential and much celebrated book about this very issue by economist Arthur Okun, 'Equality and Efficiency: The Big Trade-off' thoroughly discusses the trade-offs between efficiency and equality (Okun 1975). With his famous "leaky bucket experiment" in which he asserts that a dollar transferred from rich to poor people will result in less than a dollar increase in income of the recipient, Okun (p. 120) states "... the conflict between equality and economic efficiency is inescapable". According to Okun, transfer incurs administrative costs of redistribution, changes in savings and investment behavior induced by redistribution, and changes in attitude (e.g., motivation to acquire human capital, motivation to work) induced by redistribution. The end result is that efforts to achieve equality necessarily leads to a reduced level of income as a whole and less efficient use of resources. There is a large body of literature investigating the trade-off between efficiency and equality, for example, a general economic equilibrium analysis of the total welfare cost of the United States tax system done by Ballard et al. (1985), labor supply response to earned income tax credit (Eissa and Liebman 1996) tax avoidance and the deadweight loss of the income tax (Feldstein 1999), the impact of the (Katz and Meyer 1990) potential duration of unemployment benefits on the duration of unemployment (Katz and Meyer 1990; Meyer 1995) welfare, the earned income tax credit, and the labor supply of single mothers (Meyer and Rosenbaum 2001).

The basic economic models break down if trade-offs do not occur since the world is really full of trade-offs. If we want more of something, we have to sacrifice other things. Harvard economist Gregory Mankiw, a former White House economic adviser, concludes in his book, "Principles of Economics," that economic principles alone cannot resolve the conflict between efficiency and equity. He contends that political philosophy plays an important role as well, in striking a balance between these two goals. Unfortunately, there is not much literature in this line of inquiry. In their recent work, Berg and Ostry (2011) discover that when growth is looked at over the long term, the trade-off between efficiency and equality may not exist. They actually claim that equality seems to be an important element in promoting and sustaining economic growth. Their results suggest that the level of inequality is the main cause underlying the difference between countries that can achieve rapid growth for many years or even decades and the many others that see growth spurts fade quickly. Thus, improving equality may also improve efficiency in achieving a rapid growth in the long term. More importantly, the literature investigates the issue at an individual level. However, we investigate the transfer issue at a more aggregate level, that is, from a municipality level to another municipality level. In such a case, the administrative costs and the disincentive effects of transfer might be less. Our work investigates the issue in this important aspect.

The intergovernmental income transfer system is aimed at equalizing the provision of locally financed welfare services between municipalities. Nevertheless, there are in fact mixed results regarding the effects of increased grants on local governmental spending. Some studies conclude that net receivers will increase local governmental spending thus increasing local taxes, while other studies find that increased local grants lead to an equiproportional decrease in local taxes and thereby a corresponding increase in private incomes and spending (Brooks 2008; Singhal 2008; Lutz 2010; Litschig 2012). In a study of increased grants to some rural Finish municipalities, Lundqvist found that increased grants caused both higher local governmental spending and lower local income taxes in the short term. However, the effects on spending were of a larger magnitude than the effect on the local tax level. The long-term effects did not fully counteract the short-term effects, implying that increased grants resulted in permanent increased local governmental spending (Lundqvist 2013). It is not necessarily a one-to-one relationship between increased grants to local Authorities and an increase in local governmental spending on welfare services and decreased local taxes. Niskanen (1971),

Romer and Rosenthal (1979), Brollo et al. (2013), argue that local politicians and local civil servants could spend part of their grants to expand local bureaucracy or on corruption. There is an ongoing and unsolvable conflict between local autonomy and national equality. Some political issues are more efficiently solved at a local level, but different local preferences and resources give rise to unequal local governmental spending and welfare. National grants and income equalization measures, aimed at redistributing wealth among municipalities, have been implemented in most countries (Kearns and Forrest 2000; Berthier 2005). In the Scandinavian welfare states, this conflict is substantial due to the high ambitions regarding national equality and the decentralized, multileveled political structure (Karlsson 2015).

This topic is also equally important from a regional development and welfare perspective, which are key issues in public policy. Efficiency and equality has always been a topic of concern to public policy scholars. As the IES, which involves both equality and efficiency, is a significant economic, political and social issue on how public policies are made, this paper makes several theoretical and empirical contributions to the literature on public policy.

Our research aim is as follows: estimate the equalizing effects of IES. First, we present the formal model for the IES based on the verbal guidelines outlined by the Norwegian Department of Local Government and Modernisation. In the subsection Data and Methods, we estimate the equalizing effects of IES by comparing the tax revenues prior to IES in Tana Municipality with the post-IES tax revenue. In the subsection Results and Discussion, we provide an estimate of the equalizing effects of IES by comparing the distribution of municipal tax revenues prior to IES with the post-IES tax revenues. We conclude the study with a discussion on how the IES affects equality in terms of municipalities' incomes and their efficiency.

We expect that this paper will help identify measures that minimize costs of a policy aiming at equal distribution of income and opportunity in the public policy debate. This paper should thus make a significant contribution to theory and practice of public policy.

## 2. The Model

The model is a mathematical approach to the verbal guidelines outlined by the Norwegian Department of Local Government and Modernisation in a White paper to the Parliament. The IES consists of three components: (i) symmetric correction of tax revenues; (ii) supplemental compensation; and (iii) financing contribution. We describe each of these components and subsequently analyze the interrelationship between changes to municipality revenues before and after the IES in more detail below.

### 2.1. Symmetric Part

If $(\bar{s})$ is the average tax revenue per inhabitant of municipalities in the country as a whole, $(s_i)$ is the tax revenue of a municipality $i$ and $h$ is a specific percentage rate to be decided by the government, then the amount per inhabitant that the municipality is granted $(r_i)$ as a symmetric part (SP) of the IES is given by:

$$r_i = (\bar{s} - s_i) \cdot h. \tag{1}$$

The rate $h$ is equal to 60 percent. Clearly, $r_i > 0$ if $(\bar{s} > s_i)$ and $r_i < 0$ if $(\bar{s} < s_i)$ meaning that a municipality can be a contributor to the system if its tax revenue per inhabitant is more than the national average tax revenue per inhabitant of municipalities; otherwise, the municipality can be a recipient. The SP is a zero sum because the sum of all the incoming payments is equal to the sum of all the outgoing payments.

### 2.2. Supplemental Compensation

Municipalities with low tax revenue receive another grant called supplemental compensation (SC). The SC to a municipality $i$ per inhabitant $k_i$ is calculated as:

$$k_i = (\bar{s} \cdot g - s_i) \cdot j \quad , \forall s_i < \bar{s} \cdot g, \tag{2}$$

where $g$ and $j$, which are also decided by the government, are 90 and 35 percent, respectively, at present. This means that municipalities with tax revenue less than 90 percent of the national average receiving SC. SC is 35 percent of the difference between $0.9\bar{s}$ and the municipality's own tax revenue per inhabitant $s_i$ at present. Unlike SP, SC does not require incoming and outgoing payments to be balanced.

*2.3. Financing Contribution*

The aim of introducing financing contribution (FC) in the IES is to finance the SC. SC (expression 2) is computed for the municipalities having tax revenue per inhabitant less than 90 percent of the national average and summed for the entire country. FC per inhabitant is equal to this sum divided by the total population in the country. All municipalities pay the amount equal to FC per inhabitant multiplied by the number of inhabitants in the municipality. In 2017, the total supplementary compensation (SC) paid to the municipalities in Norway was NOK 2,022,911,463 while the population was 5,258,317. This means that the financing contribution per inhabitant in the municipality (FC) in this year was: 2,022,911,463/5,258,317 = 384.71. (NOK stands for Norwegian krones and 1 US$ = 7.78 NOK as of 18 February 2018.) To make the notation easier, we set $b = FC$ in all formulas below.

All municipalities must pay FP—including those with tax revenue per inhabitant less than 90 percent of the national average. Depending on the amount that a municipality receives as SC and pays for FC, the municipality can be a net contributor or net recipient to/from SC. A municipality is a net recipient of SC if:

$$(g\bar{s} - s_i)j - b > 0. \tag{3}$$

By rearranging and substituting $p = s_i/\bar{s}$, we get:

$$p < g - \frac{b}{j\bar{s}}. \tag{4}$$

If we use figures from 2017, we get:

$$p < 0.9 - \frac{384.71}{0.35 \cdot 29779} = 0.863. \tag{5}$$

This shows that only municipalities with tax revenue per habitant less than 86.3 percent of the national average are net recipients of the supplemental compensation.

Combining the above three components gives an expression for the amount that municipality $i$ receives through the IES per capita:

$$t_i(s_i, \bar{s}) = \begin{cases} [(\bar{s} - s_i)h - b] & , \forall s_i > g\bar{s}, \\ [(\bar{s} - s_i)h + (g\bar{s} - s_i)j - b] & , \forall s_i < g\bar{s}, \end{cases} \tag{6}$$

where $b$ denotes for financial contribution which was NOK 384.71 per inhabitant in 2017. Clearly, this expression is a function of two variables, viz., the municipality's own tax revenue $s_i$ and average tax revenue of municipalities in the country $\bar{s}$.

We can now obtain per capita net tax revenue of a municipality after adjustment prior to IES by adding the per capita tax revenue of the municipality and the correction term given in Equation (6) as:

$$i = s_i + t_i(s_i, \bar{s}) = \underbrace{s_i}_{\text{Per cap. tax}} + \begin{cases} [(\bar{s} - s_i)h - b] & , \forall s_i > g\bar{s}, \\ \underbrace{[(\bar{s} - s_i)h + (g\bar{s} - s_i)j - b]}_{\text{Correction term}} & , \forall s_i < g\bar{s}. \end{cases} \tag{7}$$

The expression (7) shows that the net per capita tax revenue of a municipality after the adjustment prior to IES depends on:

1. The average tax revenue per inhabitant in the municipality $s_i$.
2. The average tax revenue per inhabitant in the country $\bar{s}$.

To predict how a municipality's net tax revenue will get affected due to a change in the average tax revenue per inhabitant, we take partial derivative of Equation (7) with respect to the municipality's tax revenue $s_i$:

$$i'_{s_i} \approx \begin{cases} (1-h) & , \forall s_i > g\bar{s}, \\ 1-(h+j) & , \forall s_i < g\bar{s}. \end{cases} \tag{8}$$

The expression $i'_{s_i}$ thus measures a change in tax revenue per inhabitant in a municipality when the tax revenue of the municipality changes for some reason.

## 3. Data and Methods

We used data on tax revenue per inhabitant in 2017 for all 426 municipalities in Norway. Table 1 presents the descriptive statistics of tax revenue in selected municipalities. The table also shows how the tax revenue is distributed among the municipalities with and without the IES.

We applied the data to the model of the IES developed above to illustrate the model and simulate the implications. We illustrate the whole procedure and calculation for Tana municipality[1] with applicable figures from 2017 as an example. The national average municipal tax revenue per inhabitant in 2017 was $\bar{s} = 29{,}779$ NOK. The corresponding figure for the municipality of Tana was $s_b = 23{,}245$, which was 78 percent of the national average. Since $s_b < 0.9\bar{s}$, the expression for the amount that Tana received per capita from IES in 2017 is:

$$t_i(s_i, \bar{s}) = [\underbrace{(\bar{s} - s_i)0.6}_{1} + \underbrace{(0.9\bar{s} - s_i)0.35}_{2} - \underbrace{384.71}_{3}], \tag{9}$$

where term 1 is SP, term 2 is SC and term 3 is FC. If we substitute the relevant figures for taxes and population, we get:

$$t_i(s_i, \bar{s}) = [\underbrace{(29779 - 23245)0.6}_{1} + \underbrace{(0.9 \cdot 29779 - 23245)0.35}_{2} - \underbrace{384.71}_{3}] = 4780. \tag{10}$$

Tana received 4780 NOK per inhabitant in 2017 from the IES because the tax revenue per inhabitant in the municipality was significantly less than the national average.

We calculated symmetric, supplemental and financing components of the IES for each municipality using the same procedures. We also calculated the amount that a municipality could receive from or had to contribute prior to the IES. Finally, we calculated the net tax revenue of each municipality with the IES.

In addition, we estimated the effects of a marginal change in tax revenue per inhabitant of a municipality prior to the IES on the municipality's per capita net tax revenue by substituting figures as of 2017 in Equation (8) as:

$$i'_{s_i} \approx \begin{cases} (1-h) = (1-0.6) = 0.4 & , \forall s_i > g\bar{s}, \\ (1-(h+j)) = (1-(0.6+0.35)) = 0.05 & , \forall s_i < g\bar{s}. \end{cases} \tag{11}$$

---

[1] http://www.Tana.kommune.no/.

We calculated the standard deviation, range and proportion of tax revenue based on each municipality's national average to investigate the dispersion of tax revenue.

**Table 1.** Some results of the income equalization system per capita. Note: n.a. = not applicable. (Own estimates based on national data from the Government's White paper 2018 (Regjeringen 2018)).

|  | Before IES | SP | SC | FC | Total | After IES |
|---|---|---|---|---|---|---|
| Norway , average | 29,754 | 0 | 385 | −384 | 0 | 29,754 |
| Tana municipality | 23,245 | 3920 | 1245 | −384 | 4780 | 28,025 |
| Richest municipality | 74,956 | −27,111 | 0 | −384 | −27,495 | 47,465 |
| Poorest municipality | 17,414 | 7414 | 3283 | −384 | 10,313 | 27,727 |
| Standard deviation | 5878 | 4068 | n.a. | n.a. | n.a. | 1940 |
| Range | 57,542 | 27,118 | n.a. | n.a. | 37,809 | 19,738 |

Where IES is the income equalization system, SP is the symmetric part, SC is the supplemental compensation and FC is the financing contribution.

## 4. Results and Discussion

Table 1 presents the descriptive statistics of tax revenue per inhabitant including results of the model in selected municipalities. First, we present the distribution of tax revenue without the IES. Tax revenue of municipalities ranged from 17,414 to 79,956 NOK with a national average of 29,754. The tax revenue in the richest municipality was 4.3 and 2.5 times more than that of the poorest and national average municipal tax revenue, respectively. Similarly, the tax revenue in the poorest municipality was only 58 percent times of the national average. Eighty-three percent of the municipalities were net receivers while the rest, 17 percent, were contributors.

Using Equation (11), we show the effects of an increase in tax revenue per inhabitant of a municipality by NOK 1000 prior to the IES as follows:

1. A municipality at a point of departure with tax revenue more than the national average is a net contributor to the income equalisation system. If the tax revenue of the municipality increases by NOK 1000 per inhabitant, the municipality will have to increase its contribution by NOK 600 per inhabitant and it will be left with NOK 400 per inhabitant.
2. A municipality at a point of departure with tax revenue less than the national average, but more than 90 percent of the average, is a net recipient from the IES. If such a municipality increases its tax revenue, it will receive reduced transfer payments from the IES. If the tax revenue increases by NOK 1000 per inhabitant, the municipality will have to increase its contribution by NOK 600. The net increase in the municipal revenue will be NOK 400 per inhabitant (the same as in point 1 above).
3. A municipality at a point of departure having tax revenue less than 90 percent of the national average is a net recipient from the IES and receives payments from both the symmetric part and the supplemental part. If the municipality increases its tax revenue, it will receive reduced transfer payments from this system. If the tax revenue increases by NOK 1000 per inhabitant, the municipality will have reduced transfers of NOK 950. The net increase in the municipality's revenues will thus be NOK 50 per inhabitant. This indicates that poorer municipalities do not get an incentive to undertake measures to develop and extend businesses in the municipality in order to increase tax revenue.

The result is fairly robust with respect to a change in tax revenue per inhabitant of municipalities. Although the IES has been formulated in such a way that municipalities with tax revenue per inhabitant of less than 90 percent of the national average should receive supplemental compensation, only the municipalities with tax revenue less than 86.3 percent of the national average receive more from the supplemental compensation. This is because all municipalities, also those that receive supplemental compensation, contribute to financing the supplemental part in equal amounts per inhabitant. In 2017,

83 percent of the municipalities had revenue less than 86.3 percent of the national average and were thus net recipients of supplemental compensation. However, 3 percent of municipalities had tax revenue between 86.3 percent and 90 percent of the national average; they received supplemental compensation, and were net contributors to the IES. Eighty-six percent of the municipalities had less than 90 percent of the national average.

As can be seen from Figure 1, the IES prevents a significant reduction in net tax revenue of a municipality. At the same time, the poorer municipalities have less incentive to increase their tax revenue on their own. A municipality with tax revenue per inhabitant less than 90 percent of the national average will only increase its revenue by NOK 50 after the IES if its tax revenue prior to equalization increases by NOK 1000. This gives less incentives to poorer municipalities to look for ways in which to develop business.

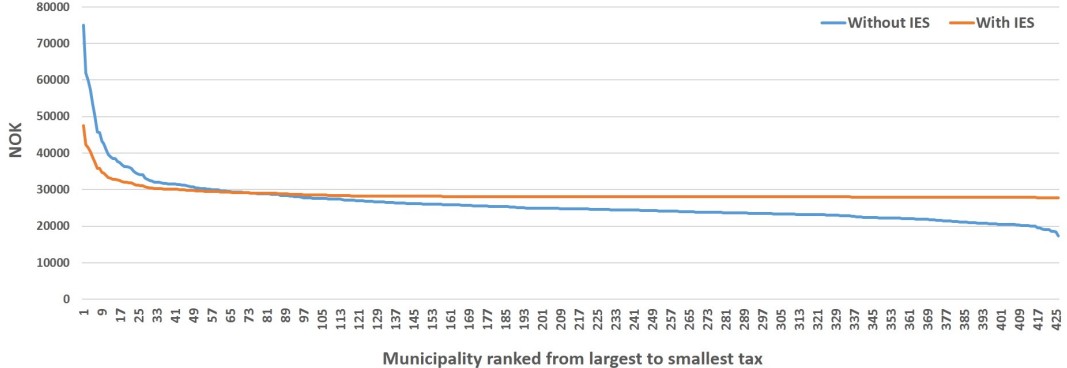

**Figure 1.** The municipalities' tax revenues before and after the income equalization system.

In general, the IES helps to significantly reduce inequality of tax revenue per inhabitant among municipalities. It also helps to stabilize revenue of municipalities. It reduces fluctuations in municipal revenues resulting from fluctuations in gross tax revenues. The poorer municipalities in particular are well-protected against large fluctuations in tax revenue. However, the scheme can cause poor municipalities to have diminished incentives to increase their tax revenue and/or accommodate business development on their own, indicating that the IES involves a trade-off between efficiency and equality. It is therefore important to implement measures that would augment equality while incurring the smallest possible loss of efficiency of the scheme.

Now, we illustrate and discuss welfare implications of the IES based on different schools of thought, simple logic, common sense, and evidence.

This analysis is based on a diminishing marginal social welfare function and a positive correlation between local tax revenue prior to IES and the wealth of the inhabitants. The principle of diminishing marginal social welfare suggests that poorer municipalities will gain more welfare from additional spending than the richer ones. The aggregate welfare of the society will be maximized by transferring some revenue from rich to poor municipalities because a loss of welfare of the rich municipality is smaller than the gain of welfare of the poor municipality. The transfer of some tax revenue from the richer to the poorer municipalities increases social welfare also according to Rawlsian theory (Rawls 1971). Such transfer also leads to distributive efficiency as per Lerner's concept of distributive efficiency because distributive efficiency occurs in a society when goods and services are received by those who have the greatest need for them (Lerner 1944).

Inequality can have several social costs such as crime, theft, riots, strikes, and so on. Despite extensive literature dealing with the efficiency costs of efforts/measures of income transfers from the rich to the poor, researchers have paid little attention to the social cost of inequality, benefits of the equality of income, opportunity in the long term and measures to minimize the costs and augment the benefits. The IES can benefit a society by increasing social efficiency at the possible cost of economic

efficiency. Joseph Stiglitz, who severely criticizes inequality, thoroughly discusses the consequences of inequality in his recent book entitled *The Price of Inequality* (Stiglitz 2012).

This issue of the IES necessarily involves political philosophy, not only economics. Inequality essentially involves our social and political values and normative judgement. A government wants to create a society that is just and fair through reasonably equal distribution of income and opportunity. The IES can help to achieve this.

The IES works in such a way that transfer from the rich to the poor municipalities is less distorting because the scheme does not reduce a person's decision to supply labor and has less impact on individuals' saving and investment behaviors. On the other hand, Okun's "leaky bucket" and inefficiency due to transfer from the rich to the poor are based on the assumption of a perfect market, but, in fact, the market itself is not perfectly competitive so the "leak" may not be so big.

Transfer is not an all-or-nothing good. People might be willing to trade a bit of their income for something they think they value more: reasonably equal distribution of income and wealth, self-respect, and so on. We decrease efficiency, but (hopefully) gain something that we would not have if there were no transfers. Sometimes, e.g., we are willing to allow the country to run a little less efficiently by a policy such as the IES to improve the living standards of those in need. We lose efficiency in exchange for reduced migration to and congestion in cities; the long-term effects could outweigh the loss of output due to transfer. Should we not be willing to lose some efficiency if this will make the country a safer, better and/or fairer place overall? We may not like it because we would certainly prefer to have extra safety without giving up anything but in the real world this is not possible; most of us are willing to make that compromise. Every decision involves a trade-off.

It is a fact that Nordic countries such as Norway, Denmark, Sweden and Finland, which pursue egalitarian policies, have maintained significantly high prosperity for so many years. These countries are also excellent examples of preserving the natural environment, cultural heritage, less pollution in the cities, controlled migration and high rankings in human development indicators. In Norway in particular, why do citizens in poor municipalities need to move to other municipalities or cities if they enjoy the same living standards where they reside at present? Norway implements a policy that makes it attractive to live in rural municipalities. The Norwegian authorities do not want the majority of people to live in or close to Oslo.

Such transfer results in an increase in aggregate welfare. However, it is indeed important to identify and implement measures that would augment equality while incurring the smallest possible loss of efficiency of such a policy.

## 5. Conclusions

The IES, which significantly reduces inequality of municipal tax revenue over time and space, helps to stabilize revenues across municipalities over time. It reduces fluctuations in municipal revenues resulting from fluctuations in gross tax revenue. Poorer municipalities, which are protected against a significant reduction in their tax revenue, can maintain their welfare programs and services. This, in turn, helps not only to stabilize the welfare of citizens in a municipality over time, but also to maintain similar welfare standards across municipalities. As a result, the aggregate welfare in the nation increases. However, the IES can cause poor municipalities to have diminished incentives to increase their tax revenue and/or accommodate business development on their own, indicating that the scheme involves a trade-off between efficiency, economic efficiency in particular, and equality. It is therefore important to implement measures that would augment equality while incurring the smallest possible loss of efficiency.

**Author Contributions:** All authors contributed equally to this work.

**Conflicts of Interest:** The authors declare no conflict of interest.

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
