# Peer review of "The Income Equalization System among Municipalities in Norway: Strengths and Implications"

_economies, doi:10.3390/economies6020034_

Round 1

Reviewer 1 Report

A lot of assertions must be substantiated.

An English language editing service must be used.

A lot of sentences are way too long. E.g. “A highly influential and much celebrated book about this very issue by economist Arthur Okun, ‘Equality and Efficiency: The Big Tradeoff’” should be “Okun (1975).” Or “Harvard economist GregoryMankiw, a former White House economic adviser, concludes in his book, ‘Principles of Economics.’” Books titles must be written in italics.

The figures must specify the sources.

The list of references is too short and most titles are too old.

Author Response

Comment: An English language editing service must be used. A lot of sentences are too long.

Answer: The paper has been corrected by a native-english speaking academic.

Comment: The list of references is too short and most titles are too old. 

Answer: Corrected. 12 additional references have been added to the list.  The subsection called “Introduction” has been substantially expended. 

Reviewer 2 Report

Major concerns

-At first it must be noted that the text is full of questions marks, which make the text very hard to read and the English has to be improved.

-Neither in the abstract nor in the introduction the authors make clear what their research objective is and from where they got their formulas? Are the latter taken from laws directly, or have the authors derived them from laws on their own? Where can I find the source for the formulas?

-They authors argue that a redistribution of the tax revenue between municipalities, is in some way the same as the redistribution of income between rich and poor citizens. However, it cannot be excluded that this analogy could be correct, but in general I fear that this interpretation is misleading, because it is by far not guaranteed that the poor are better off with, than the without the IES. To make the point clear, the usual argument against such a redistribution of tax revenue is that mayors, who are able to attract firms, to organize efficiently the local administration and to increase in this way the tax revenue have to transfer part of their efforts’ baenefits to mayors, who may waste the money for inefficient projects. Accordingly, the latter have no incentive to change their behavior or the objective to increase the efficiency of their administration. However, if the authors think that poor people and not the middle class benefit from the IES, then they have to provide evidence that this is true.

-On page 1, the authors argue, ‘In this paper, we attempt in investigating those circumstances under which there is not much trade-off between equality and efficiency, at least in the long term.’, but in the whole paper the authors do not investigate in the costs of collecting the taxes and associated deadweight losses caused by taxation, and therefore the authors do not know anything about the extent of the trade-off and how inefficient the IES is.  

-In general the authors make a lot of statements, and then do not provide a reference or alternative evidence. For example, ‘Economists are also blamed in emphasizing efficiency, economic efficiency in particular, at the cost of equality ?.’, but who is blaming? Further it looks like as if economists are not aware of the income distribution’s importance (except A. Smith according the authors), but in standard text books like the ‘Principles” from Paul Krugman and Robin Wells the importance of income and wealth inequality is well recognized as an important public good. Of course libertarians like Friederich Hayek or Milton Friedman have a different view, but they are not representative for all economists. However, it has to be accepted as truth, that taxes create deadweight losses, otherwise it is hardly to explain, why many people, particularly rich people try to evade taxes and commit tax fraud (Are there no Norwegian citizens in the Panama or Paradise papers?). In so far the authors’ statement (page 1), ‘It is generally accepted that if there is an opportunity to transfer a small fraction of our income to someone who needs it more than we, we should not hesitate.’, is wrong. What is a small share of income? As I worked in the Netherlands, my tax rate including social security contribution, health insurance and unemployment insurance added up to nearly 50%. Is that a small share? For me it was okay, but I know many who think that is by far too much.

-One main problem for the reader is that she has no idea how the tax revenue of municipalities is generated (what kind of taxes? Particularly have the municipalities the right to raise local taxes, or are all taxes and tax rates are determined by the central of regional government?). It is important to provide much more information about the tax system in Norway, before the redistribution of tax revenues can be discussed.

-It may surprise the authors, but many European countries apply similar redistribution mechanisms between municipalities or regions. For example Germany, Italy or Spain apply similar mechanisms. Partly the respective citizens of the rich regions and cities perceive this redistribution as unfair, because they argue, that they have to finance the inefficiency of bad governed regions and cities. I would wonder if this is different in Norway. In Germany, in the course of reunification the Western part transferred in total around 2 trillion EUR to the former German Democratic Republic, and now partly the cities, which are net contributors have a worse public infrastructure than the net benefitting cities in East Germany. Another extreme result of this redistribution could be observed in Spain last year, where many supporters of the Catalan independence movement are motivated by the fact that Catalonia has to transfer a huge share of its tax revenue to the Southern regions in Spain. The right-wing political parties in Italy are strongest in the rich North of Italy, because they have to finance the poor South of Italy. In so far such a system is not unique but widely spread in Europe and mostly associated with some form of conflict. Regarding this issue, the authors do not cite one paper of regional fiscal equalization, although this is an important issue in regional economics.

-Another point is that the authors write (Page 9), ‘We used data on tax revenue per inhabitant in 2017 for all 426 municipalities in Norway.’ which is nice, but where is the source of the data?

-In the section 4 and 5 the authors claim repeatedly that the IES leads to an increase in national welfare. That raises the question what is ‘national welfare’ in the view of the authors, welfare in the sense of Rawls’ ‘’Theory of Justice’ or for example in the sense of Jeremy Bentham (1776). ‘A Fragment on Government’. And even if the authors have answered this question, it remains to proof that the claim is true. Again, how do the authors know, that mainly poor people benefit from the fiscal equalization? Or do the authors assume rich people live in rich cities, poor people live in poor cities?

Recommendation how to improve this paper:

In principle, I like the topic of the paper, but in the present form I doubt that it can be published. Thus, two approaches came into my mind how to explore the idea of the authors to create a publishable paper.

1. Either the authors set up a regional model with two municipalities. Then they can analyze what will be the theoretical consequences of redistribution of tax revenue with regard to the poor in both cities and compare it with real numbers.

or

2. They compare fiscal equalization between municipalities/regions in different countries discussing the pros and cons of the different mechanisms.

Author Response

Comment: The text is full of questions marks, which make the text very hard to read and the English has to be improved.

Answer: The paper has been corrected by a native-english speaking academic.  The question marks have been removed.

Comment: The authors should make clear what their research objective is. 

Answer: We have made some changes at the end of the subsection called “Introduction”.

 “First, we present the formal model for the IES based on the verbal guidelines outlined by the Norwegian Department of Local Government and Modernisation. In the subsection Data and methods we estimate the equalizing effects of IES by comparing the tax revenues prior to IES in Tana Municipality with the post-IES tax revenue. In the subsection Results and discussion, we provide an estimate of the equalizing effects of IES by comparing the distribution of municipal tax revenues prior to IES with the post-IES tax revenues.  A discussion of how the IES effects equality in municipalities’ incomes and their efficiency wraps up the study”. 

Comment: Where do they get their formulas? Are the latter taken from laws directly, or have the authors derived them from laws on their own? Where can I find the source for the formulas?

Answer: The model is a mathematical approach to the verbal guidelines outlined by the Norwegian Department of Local Government and Modernisation in a White paper to the Parliament. The source of this paper is: https://www.regjeringen.no/no/tema/kommuner-og-regioner/kommuneokonomi/gront-hefte/id547024/. (In Norwegian).

Comment: They authors argue that a redistribution of the tax revenue between municipalities, is in some way the same as the redistribution of income between rich and poor citizens. However, it cannot be excluded that this analogy could be correct, but in general I fear that this interpretation is misleading, because it is by far not guaranteed that the poor are better off with, than the without the IES. To make the point clear, the usual argument against such a redistribution of tax revenue is that mayors, who are able to attract firms, to organize efficiently the local administration and to increase in this way the tax revenue have to transfer part of their efforts’ benefits to mayors, who may waste the money for inefficient projects. Accordingly, the latter have no incentive to change their behavior or the objective to increase the efficiency of their administration. However, if the authors think that poor people and not the middle class benefit from the IES, then they have to provide evidence that this is true.

Answer:

We do agree. We have therefore added the following text to the introduction:  “It is not necessarily a one-to-one relationship between increased grants to local Authorities and an increase in local government spending on welfare services and decreased local taxes. Niskanen (1971), Romer and Rosenthal, 1979 and Brollo et al, 2013 argue that local politicians and local civil servants could spend part of their grants to expand local bureaucracy or on corruption”.   

Comment: On page 1, the authors argue, ‘In this paper, we attempt in investigating those circumstances under which there is not much trade-off between equality and efciency, at least in the long term.’, but in the whole paper the authors do not investigate in the costs of collecting the taxes and associated deadweight losses caused by taxation, and therefore the authors do not know anything about the extent of the trade-off and how inefficient the IES is.

Answer: Agree. That sentence has been erased from the paper.

Comment: It has to be accepted as truth, that taxes create deadweight losses, otherwise it is hardly to explain, why many people, particularly rich people try to evade taxes and commit tax fraud (Are there no Norwegian citizens in the Panama or Paradise papers?)

Answer: There is a consensus among economists that taxes, almost without any exceptions, create a deadweight loss. It is also true that some Norwegians have committed tax frauds and some of them were mentioned in the Panama papers. This is not, however, a proof of a deadweight loss. Even in the case of a polltax, individuals and firms would have incentives for tax evasion and would probably spend effort and financial resources to commit tax crimes. 

Comment: In so far such a system is not unique but widely spread in Europe and mostly associated with some form of conflict. Regarding this issue, the authors do not cite one paper of regional fiscal equalization, although this is an important issue in regional economics.

Answer: Agree. We have added some references describing the IES in other countries; Lundqvist (Finnish), Karlsson (swedish), Kearns & Forrest and Berthier (several countries).

Comment: (Page 9), ‘We used data on tax revenue per inhabitant in 2017 for all 426 municipalities in Norway.’ Where is the source of the data?

Answer: We have added a reference attached to table 1.  “Own estimates based on national data from the Governments White paper (Regjeringen, 2018)”. 

Comment: In the section 4 and 5 the authors claim repeatedly that the IES leads to an increase in national welfare. That raises the question what is ‘national welfare’ in the view of the authors, welfare in the sense of Rawls’ ‘’Theory of Justice’ or for example in the sense of Jeremy Bentham (1776). ‘A Fragment on Government’. And even if the authors have answered this question, it remains to proof that the claim is true. Again, how do the authors know, that mainly poor people benefit from the fiscal equalization? Or do the authors assume rich people live in rich cities, poor people live in poor cities?

Answer: We have changed the sentence in line 165 to clarify that we assume a diminishing marginal social welfare function and that there is a positive correlation between local tax revenue prior the wealth of the inhabitants.

 “This analyses is based on a diminishing marginal social welfare function and a positive correlation between local tax revenue prior to IES and the wealth of the inhabitants”.